# Undernutrition and its determinants among adolescent girls in low land area of Southern Ethiopia

Yoseph Halala Handiso[1,2]* , Tefera Belachew[3], Cherinet Abuye[4],
Abdulhalik Workicho[5], Kaleab Baye[2]

1 School of Public Health, College of Health Sciences and Medicine, Wolaita Sodo University, Sodo, Ethiopia, 2 Center for Food Science and Nutrition, College of Natural and Computational Sciences, Addis Ababa University, Addis Ababa, Ethiopia, 3 Human Nutrition Unit, College of Public Health & Medical Science, Jimma University, Jimma, Ethiopia, 4 Save the Children, Addis Ababa, Ethiopia, 5 Friedman School of Nutrition Science and Policy, Tufts University, Boston, Massachusetts, United States of America

ʘ These authors contributed equally to this work.
* yosefhalala@gmail.com

## Abstract

### Background

Undernutrition is one of the most common causes of morbidity and mortality among adolescent girls worldwide, especially in South-East Asia and Africa. Even though adolescence is a window of opportunity to break the intergenerational cycle of undernutrition, adolescent girls are a neglected group. The objective of this study was to assess the nutritional status and associated factors among adolescent girls in the Wolaita and Hadiya zones of Southern Ethiopia.

### Methods

A community-based cross-sectional study was conducted, and a multistage random sampling method was used to select a sample of **843** adolescent girls. Anthropometric measurements were collected from all participants and entered in the WHO Anthro plus software for Z-score analysis. The data was analyzed using EPI-data 4.4.2 and SPSS version 21.0. The odds ratios for logistic regression along with a 95% confidence interval (CI) were generated. A P-value < 0.05 was declared as the level of statistical significance.

### Result

Thinness (27.5%) and stunting (8.8%) are found to be public health problems in the study area. Age [AOR(adjusted odds ratio) (95% CI) = 2.91 (2.03–4.173)], large family size [AOR (95% CI) = 1.63(1.105–2.396)], low monthly income [AOR (95% CI) = 2.54(1.66–3.87)], not taking deworming tablets [AOR (95% CI) = 1.56(1.11–21)], low educational status of the father [AOR (95% CI) = 2.45(1.02–5.86)], the source of food for the family only from market [AOR (95% CI) = 5.14(2.1–12.8)], not visited by health extension workers [AOR (95% CI) = 1.72(1.7–2.4)], and not washing hand with soap before eating and after using the toilet

**Data Availability Statement:** All relevant data are within the manuscript and its Supporting Information files.

**Funding:** Addis Ababa University Center of Food Science and Nutrition, Wolaita Sodo University and Tufts University have supported small research grant.

**Competing interests:** All authors have declared that no competing interests exist.

[AOR (95% CI) = 2.25(1.079–4.675)] were positively associated with poor nutritional status of adolescent girls in the Wolaita and Hadiya zones, Southern Ethiopia.

## Conclusion

Thinness and stunting were found to be high in the study area. Age, family size, monthly household income, regularly skipping meals, fathers' educational status, visits by health extension workers, and nutrition services decision-making are the main predictors of thinness. Hand washing practice, visits by health extension workers, and nutrition services decision-making are the main predictors of stunting among adolescent girls. Multisectoral community-based, adolescent health and nutrition programs should be implemented.

## Background

Adolescence is defined by WHO as the age range from 10–19 years, and it is a period of transition from childhood to adulthood [1]. The adolescent age group comprises 20% of the global population [2]. Malnutrition, particularly undernutrition, is highly prevalent among adolescents in low and middle-income countries [3, 4]. Nutrition status among adolescents is an important determinant of health outcomes; undernutrition affects the health status of adolescent girls. In addition to causing significant mortality, it creates long lasting effects on the growth, development, and physical fitness of survivors [5]. This, in turn, affects their ability to learn and work at maximal productivity [6]. Undernutrition is an indicator of poor nutrition and has major consequences on human health as well as the social and economic development of the population [7]. Physical growth and development during puberty increase requirements for energy, protein, and many vitamins and minerals, and deficiencies can lead to physiological, anatomical, and functional disturbances [8].

The nutritional status of adolescent girls can have intergenerational effects because adolescent girls with poor nutritional status are more likely to give birth to low birth weight infants [8, 9]. Focus on adolescent girls is important because their health and nutritional status before as well as during pregnancy influences fetal growth and newborn health. Adolescent girls' health and undernutrition is an important determinant of adverse fetal outcomes, including low birth weight, preterm births, stillbirths, and an increased risk of neonatal mortality [10]. Therefore, adequate nutrition is key; it is associated with a better quality of life and has many intergeneration benefits[11].

Most causes of malnutrition are related to poor care, poor economic status, and food insecurity; however, malnutrition can sometimes be inherited genetically [12]. Presence of malaria infections, cigarette smoking, alcohol and drug use, environmental pollution, and domestic violence are predictors of undernutrition [13]. Similarly, age of adolescent girls, occupation of father [14, 15], poor dietary diversity score, meal skipping, not getting nutrition information, living in food in-secured households [16, 17], eating less than 3 meals per day, having family size >5, source of drinking water, monthly income were predictors of under nutrition among adolescent girls [18–22].

In regions of South-East Asia and Africa, a large number of adolescent girls suffer from chronic undernutrition, which adversely impacts their own health and development, as well as that of their offspring, contributing to an intergenerational cycle of malnutrition [23, 24]. More than 10% of girls were underweight in Mauritius, Bangladesh, Maldives, Cambodia, and Vietnam [25]. Body mass index of adolescent girls were less than 20 in South Asia, Southeast

Asia, East Africa, West Africa, and Central Africa [26]. A study from northern Ethiopia reported high levels of stunting (26.5%) and thinness (58.3%) among adolescents [27].

Even though the sustainable development goals (SDGs) include an adolescent nutrition service which is addressing adolescent malnutrition, the nutritional status of adolescent girls is not improving [28]. The government of Ethiopia officially launched the National Nutrition Program (NNP) in 2009, which aimed to reduce malnutrition in Ethiopia by integrating adolescents' nutrition into community-based health and development programs but faced many challenges. The Ethiopian NNP II (2016–2020) incorporated initiatives to improve the nutritional status of adolescent girls, but these interventions are not effective [29, 30]. However, these studies were conducted among school adolescent girls. Thus, the results of these studies cannot be generalized to the whole adolescent girls. In addition to this, there are no community based studies conducted in Southern Ethiopia among adolescent girls. Therefore, understanding nutritional status and its associated factors are critical to timely address malnutrition in this age group.

## Materials and methods

### Study area

The study was conducted in the Wolaita and Hadiya zones of Southern Ethiopia. These zones are predominantly dependent on agriculture, practicing mixed crop-livestock production and living in permanent settlements. Within their landholdings, community members cultivate fruits, vegetables, roots, and tuber crops.

Fig 1 shows Map of the study sites (Wolaita and Hadiya zones) in southern nation nationality and peoples region (SNNPR), 2019. A community-based cross-sectional study was conducted at two zones in Southern Ethiopia from April 30, 2019 to May 30, 2019. The inclusion criteria were adolescent girls (both attending and not attending school) between the ages of 10–19 years in two Southern Ethiopian zones. Participants who met the inclusion criteria were randomly selected to be the study population. BMI-for-age Body mass index for age z-score and height-for-age z-score were the dependent variables. Age, educational status of the participant, family size, maternal and paternal educational level, access to nutritional counseling services in health facilities, deworming tablets, iron-folic acid supplementation, household monthly income, source of food, and number of meals per day were the independent variables for our study.

### Sample size determination

A single population proportion formula, $[n = z\frac{\alpha_2}{2} P (1-P) /d^2]$ was used to estimate the sample size. From the literature review, the prevalence of thinness (24.4%) and stunting (29.4%) were used for sample size calculations. Sample size calculation by using thinness (24.4%) was n = ($Z_{\alpha/2})^{2*} p (1-p)/d^2$ = 748 and sample size calculation by using stunting (29.4%) was n = n = ($Z_{\alpha/2})^{2*} p (1-p)/d^2$ = 843. So that for this study, stunting (29.4%) was selected to estimate the sample size as it gives a larger sample; considering a 95% confidence interval (CI) and d = 0.05%, the initial sample size was 383. By adding 10% for non-response and a design effect of 2.4, the final sample size was **843**. n = ($Z_{\alpha/2})^{2*}$p $(1-p)$ DE $/d^2$. Where: Z = Standard normal distribution value at 95% CI = $(1.96)^2$, DE = design effect, and d = 0.05 (5% margin of error).

### Sampling procedures

This study used multistage sampling techniques and was conducted in the Wolaita and Hadiya zones. From these two selected zones, two districts were selected based on a simple random sampling procedure, the Humbo district from Wolaita zone and the Misrak Badawacho district from the Hadiya zone. Three kebeles (villages) were selected from each district using a simple random

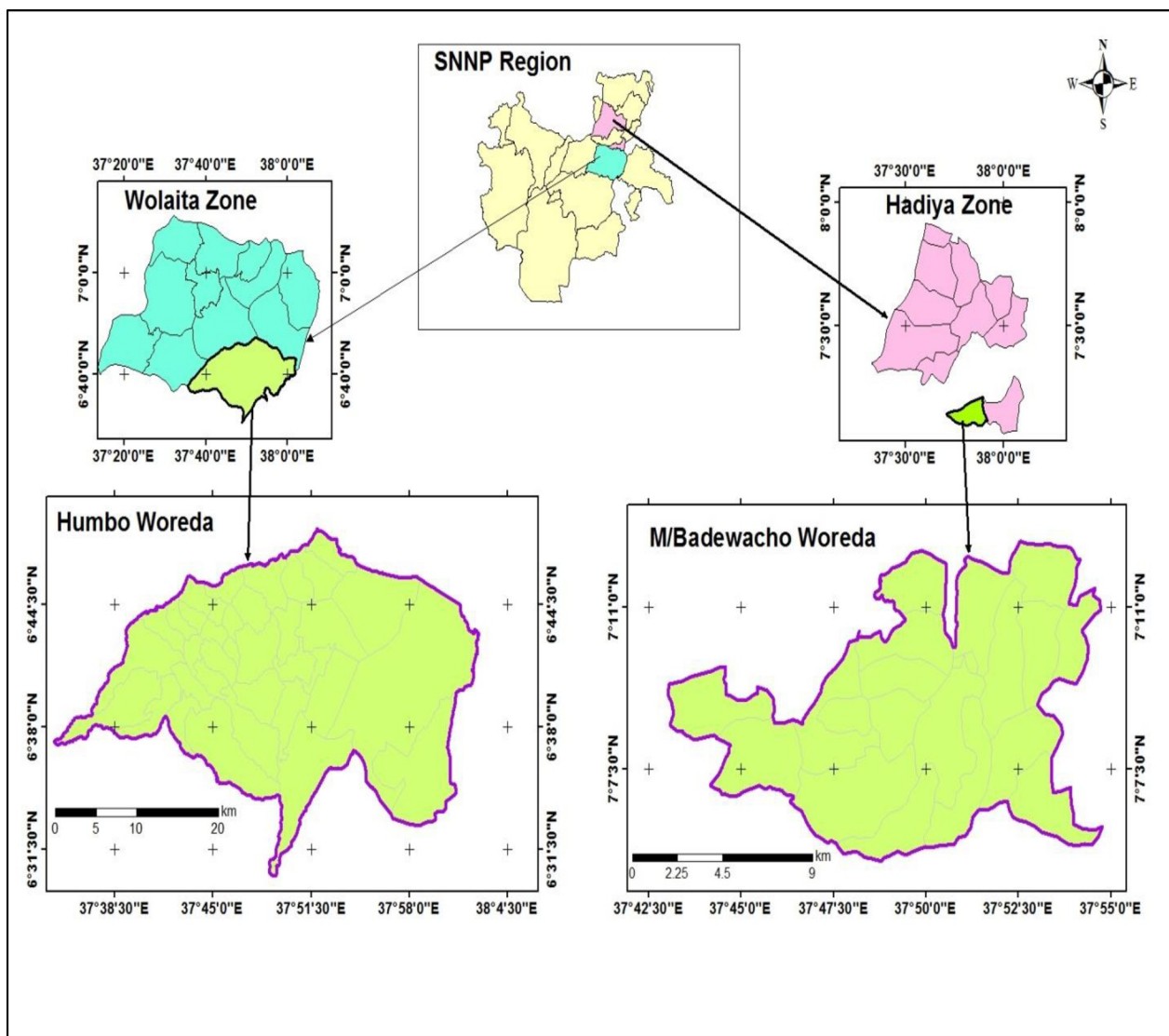

**Fig 1. Map of the study sites (Wolaita and Hadiya zones) in southern nation nationality and peoples region.**

sampling method. A listing of adolescent girls was conducted at these selected kebeles. This listing was developed with the help of both the local government administration, woreda in particular, and health extension workers. During the development of the list, if there were more than one adolescent girl in a household, one adolescent girl was selected by simple random sampling (lottery method). From the selected six kebeles, 843 participants were chosen by simple random sampling method depending on the number of adolescent girls in each kebele. Participants were drawn from each kebele based on probability proportional to size (PPS) sampling techniques. The sampling techniques depended on the number of adolescent girls in each kebele. Adolescent girls with pregnancy, physical and mental disability were excluded from the study.

## Data collection

**Anthropometric measurements.** Anthropometrics (i.e., height and weight) were measured on all sampled adolescent girls. Weight was measured to the nearest 100 g using a standard SECA

digital scale while the participants wore light clothing and no shoes. The scale was calibrated after weighing each participant. Height was measured in a standing position to the nearest 0.1 cm using a vertical board with a detachable sliding headpiece. Measuring tape was attached to it. BMI-for- age z-scores and height-for-age z-scores were calculated using the height, weight, and age of the participants. WHO Anthro plus software was used to calculate Z-score.

A structured interviewer-administered questionnaire was used to collect data. The questionnaire was developed based on a thorough review of the current literature [31–34]. A total of eight nurses with B.Sc. degrees; previous experience in collecting data; and knowledge of the culture, language, and norms of the community were employed to collect data using a pre-tested structured questionnaire. In addition to this, two supervisors with M.Sc. in public health were employed to supervise the data collection process. Data were collected on weekends for adolescent girls who attended school during the weekdays. The principal investigator controlled the daily overall study activities.

**Statistical analysis.**   First, the data were checked for completeness and consistency for data entry and cleaning. Then, data were entered into the computer using EPI-data version 4.4.2 and exported to SPSS version 21.0 for further analysis. Descriptive statistics such as frequencies, proportions, and cross-tabulation were used to present the data. In addition, bivariate logistic regression analysis was performed to assess the association between independent and dependent variables. Variables that showed an association (p-value $\leq$ 0.25) in the bivariate analysis were included in the final multivariate logistic regression model. Odds ratios for logistic regression along with a 95% CI were estimated. A p-value less than 0.05 was declared statistically significant.

**Data quality assurance.**   The questionnaire was prepared in English, translated to Amharic, and back translation to English to maintain consistency of the questions. Data collectors and supervisors were trained for 4 days to properly fill out the questionnaire and measure anthropometry. Data collectors were selected from each zone so they could communicate fluently in the local language and understand the socio-cultural practices of the community. The questionnaire was pre-tested on 5% adolescent girls in a similar area to the study sites to ensure reliability. Feedbacks from the pre-test were incorporated into the final questionnaire design. Principal investigator and supervisors performed checks on the spot and reviewed all the completed questionnaires to ensure completeness and consistency of the information collected.

Standardization of anthropometric measurements was conducted. To standardize anthropometric measurements, during training an expert took two heights and weight measurements for ten adolescent girls and then let each data collector take the measurements for all ten girls twice. Then, the averages of the two measurements for each adolescent girl taken by the data collector were compared with the average of the expert's measurements. The technical error of measurement (TEM) and coefficient of variance (CV) were computed for all data collectors using Emergency Nutrition Assessment (ENA) for SMART software. Data collators with unacceptable TEM and CV were asked to repeat the steps again.

**Ethical considerations.**   The study was approved by Addis Ababa University (AAU), College of Natural Sciences Research Ethics Review Committee. The official letter of cooperation was written to the Wolaita and Hadiya zones, and the district of health offices. The nature of the study was fully explained to the study participants and parents/guardians. Informed verbal and written consents were obtained from the parents/guardians for adolescent girls aged < 18 years old and assent was obtained from the participant before the interview. Participants $\geq$ 18 years aged were asked to provide verbal and written consent. The collected data were kept confidential. Each participant was given a code number, and the data were stored in a secure and password-protected database.

## Results

### Socio-demographic characteristics of adolescent girls in Southern Ethiopia

Eight hundred and twenty adolescent girls participated with a response rate of 97.3%.

As shown in Table 1, the mean age of the study participants was 14.6 (±1.9) years, the mean family size was 6.56 (±1.83) persons, while 69.3% of the households had ≥ 5 family members and 30.7% had < 5 family members. Most of the study participants (93.3%) were in grades 1–8 and only 0.2% had college and University education. Most of the study participants were Protestant (65.0%), but 34.3% were Orthodox Christian, and only 0.7% were Muslims. About one third (33.4%) of the study participants were from households that have less than 1000 ETB (31.25 USD) monthly income and 30.3% are from households that have greater than 2000 ETB (62.5USD) monthly income.

### Nutrition service and health-related factors of adolescent girls in Southern Ethiopia

As indicated in Table 2, 70.4% of the study participants did not receive nutrition education. Only 29.6% of the study participants had nutrition education. Similarly, 54.9% of the study participants never received deworming tablets. Out of the participants who took deworming tablets (45.1%), 65.6% have taken two tablets (albendazole,400 mg) and 34.4% have taken one tablet every six months. When considering iron and folate supplementation, only 0.4% of the study participants have taken iron-folate supplement. Of the total study participants with access to nutrition services, only 60.4% received friendly nutrition service. In 66.1% of the households, the fathers were the primary decision-makers regarding nutrition service. 27.8% of the study participants had a cough in the two weeks before data collection.

**Table 1. Socio-demographic characteristics of adolescent girls in Southern Ethiopia, 2019.**

| Variables | | Frequency(n) | Percent (%) |
|---|---|---|---|
| Age | 10–14 | 393 | 47.9 |
| | 15–19 | 427 | 52.1 |
| Median age | 14.6 ±1.9 years | | |
| Educational status | No formal education | 4 | 0.5 |
| | 1–8 grade | 765 | 93.3 |
| | 9–12 grade | 49 | 6.0 |
| | College and University | 2 | 0.2 |
| Religion | Orthodox | 281 | 34.3 |
| | Protestant | 533 | 65.0 |
| | Muslim | 6 | 0.7 |
| Family size | <5 family members | 252 | 30.7 |
| | ≥5 family members | 568 | 69.3 |
| Median family size | 6.56 ±1.83 | | |
| Monthly household income | < 1000 ETB (31.25 USD) | 274 | 33.4 |
| | 1000 ETB(31.25 USD)– 2000 ETB (62.5USD) | 298 | 36.3 |
| | > 2000 ETB (62.5USD) | 248 | 30.3 |

Source: Field survey, 2019; ETB, Ethiopian Birr

**Table 2. Nutrition service and health-related factors of adolescent girls in Southern Ethiopia, 2019.**

| Variables | | Frequency(n) | Percent (%) |
|---|---|---|---|
| Received nutrition education within the last three months | Yes | 243 | 29.6 |
| | No | 577 | 70.4 |
| Received deworming tablets every six months | Yes | 450 | 54.9 |
| | No | 370 | 45.1 |
| Number of deworming tablet received | One | 155 | 34.4 |
| | Two | 295 | 65.6 |
| Received iron folic acid supplementation (IFAS) | Yes | 3 | 0.4 |
| | No | 817 | 99.6 |
| Friendly nutrition service received | Yes | 495 | 60.4 |
| | No | 324 | 39.5 |
| Decision maker for nutrition service | Father | 542 | 66.1 |
| | Mother | 78 | 9.5 |
| | Jointly(Mother & Father) | 200 | 24.4 |
| Presence of cough within 2 weeks before data collection | Yes | 228 | 27.8 |
| | No | 592 | 72.2 |

Source: Field survey, 2019; IFAS, = Iron- folic acid supplementation

## Health and sanitation-related factors of adolescent girls in Southern Ethiopia

Table 3 describes the health and sanitation related conditions of the adolescent girls. Of the total 820 subjects, 53.0% of the adolescent girls lived in houses with mud floors, and 58.5% of the adolescent girls lived with domestic animals in the same house. Similarly, 53.3% washed their hands sometimes before eating their food, 41.7% usually washed their hands before eating, 3.4% did not wash their hands at all, 93.2% washed their hands after using the toilet, and 6.8% did not wash their hands at all after using the toilet. When washing their hands, 90.1% of the study participants reported using soap. Out of the total participants who used soap when washing their hands, only 42% usually used soap and 58% sometime used soap.

## Meal patterns of adolescent girls in Southern Ethiopia

As indicated in Table 4, 39.5% of the study participants ate $\geq$ four times per day and 59.8% of the study participants ate three times per day. This indicates only 0.7% of the study participants skipped regular meals. Similarly, 41.6% of the study participants ate smaller meals that do not satisfy their needs. Maize was the primary staple food for 40.6% of the study participants, and 38.8% consumed both teff and maize as a staple food. Participants purchased food from the market (40.4%) or produced their own food (50.6%).

## Nutritional status of adolescent girls in Southern Ethiopia

As shown in Table 5 and Fig 2, 69.5% of the adolescent girls have a normal body mass index, i.e. body mass index-for-age z-score is between -2 and +2. From the total adolescent girls, 19.5% were moderately thin as defined by a body mass index-for-age z-score is -3 $\leq$ BAZ < -2 and 8% were severely thin as defined by a BMI-for-age z-score is BAZ < -3. Only 3% of the adolescent girls were overweight (BAZ $\geq$+2).

As shown in Table 5 and Fig 3, 91.2% of adolescent girls had a normal height-for-age z-score is HAZ > -2, 7.8% were moderately stunted (-3$\leq$ HAZ<-2), and 1% were severely

**Table 3. Health and sanitation-related factors of adolescent girls in Southern Ethiopia, 2019.**

| Variables | | Frequency(n) | Percent (%) |
|---|---|---|---|
| Type of floor adolescent girls are living on | Cement | 385 | 47.0 |
| | Muddy | 435 | 53.0 |
| Living with animals in the same house. | Yes | 480 | 58.5 |
| | No | 340 | 41.5 |
| Number of windows in the entire house | 0 | 3 | 0.4 |
| | 1 | 43 | 5.2 |
| | 2 | 231 | 28.2 |
| | 3 | 297 | 36.2 |
| | 4 | 235 | 28.7 |
| | 5 | 11 | 1.3 |
| Frequency of teeth brushing (times per day) | 0 | 29 | 3.5 |
| | 1 | 401 | 48.9 |
| | 2 | 267 | 32.6 |
| | 3 | 123 | 15.0 |
| Hand washing before eating | Not at all | 32 | 3.9 |
| | sometimes | 442 | 53.9 |
| | Usually | 346 | 42.2 |
| Hand washing after using the toilet | Yes | 764 | 93.2 |
| | No | 56 | 6.8 |
| Using soap when washing hands | Yes | 739 | 90.1 |
| | No | 81 | 9.9 |
| Frequency of using soap when washing hands | Sometimes | 429 | 58.0 |
| | Usually | 310 | 42.0 |

Source: Field survey, 2019

stunted(HAZ <-3) [35]. As indicated in Figs 4 and 5, nutritional status of adolescent girls was lower than the reference population according to WHO-2007 growth chart.

## Association between variables and nutritional status of adolescent girls in Southern Ethiopia

The present study showed an association between some variables with nutritional status, as defined by BMI for age z-score (BAZ), of the study participants. Low BAZ was statistically and

**Table 4. Meal patterns of adolescent girls in Southern Ethiopia, 2019.**

| Variables | | Frequency(n) | Percent (%) |
|---|---|---|---|
| Number of meals per day | Two times | 6 | 0.7 |
| | Three times | 490 | 59.8 |
| | Four times and above | 324 | 39.5 |
| Skip regular meals | Yes | 6 | 0.7 |
| | No | 814 | 99.3 |
| Staple food | Teff | 169 | 20.6 |
| | Maize | 333 | 40.6 |
| | Teff & Maize | 318 | 38.8 |
| Source of food for the family | Produce your own | 415 | 50.6 |
| | Market purchase | 336 | 41.0 |
| | Produce your own and market purchase | 69 | 8.4 |
| Eat small meals | Yes | 341 | 41.6 |
| | No | 479 | 58.4 |

Source: Field survey, 2019

**Table 5. Nutritional status of adolescent girls in Southern Ethiopia, 2019.**

| Variables | Level | Frequency(N) | Percent (%) |
|---|---|---|---|
| Normal | -2< BAZ <+2 | 569 | 69.5 |
| Moderate thinness | -3 ≤ BAZ < -2 | 160 | 19.5 |
| Severe thinness | BAZ < -3 | 66 | 8.0 |
| Overweight | BAZ ≥+2 | 25 | 3.0 |
| Normal height | HAZ > -2 | 748 | 91.2 |
| Moderately stunted | -3 ≤ HAZ ≤ -2 | 64 | 7.8 |
| Severely stunted | HAZ <-3 | 8 | 1.0 |

Source: Field survey, 2019; BAZ, BMI-for-age z-score; HAZ, height-for-age z-score

positively associated with younger age, large family size, low monthly household income, not receiving deworming tablet(s), low educational status of the participant's fathers, separate decision making for nutrition service, source of food for family from market and not being visited by health extension workers at home (Table 6).

There was also an association between some variables with nutritional status, as defined by height-for-age z-scores (HAZ), of the study participants. Low HAZ of the study participants was statistically and positively associated with separate decision making for nutrition service, not washing hands before eating and after using the toilet, and not visited by health extension worker (Table 7).

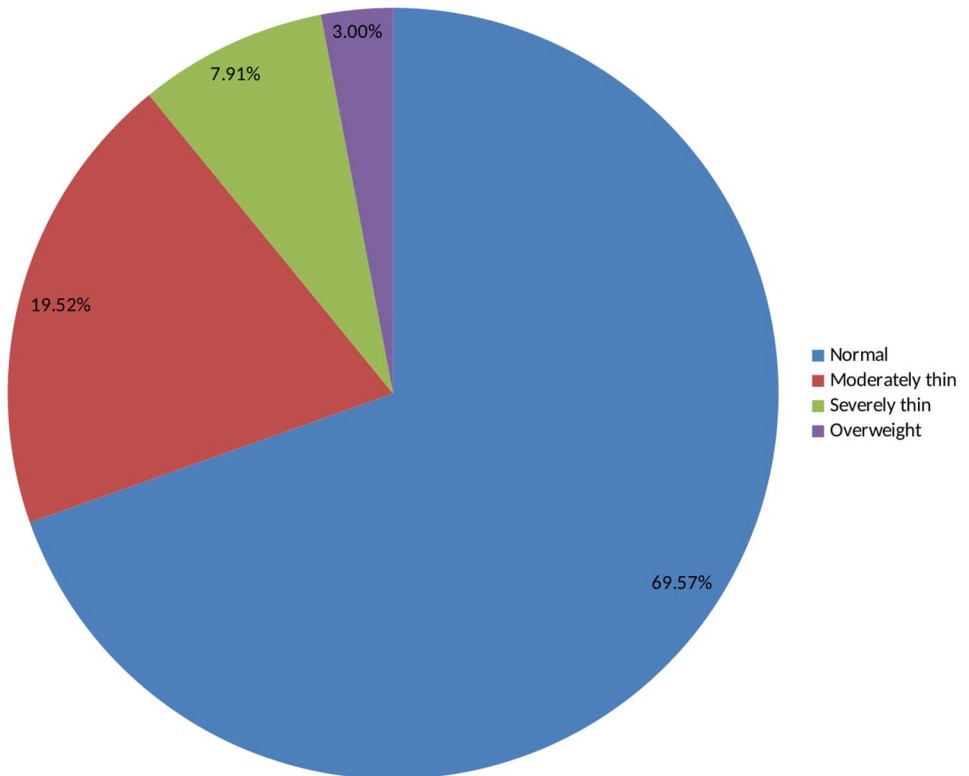

**Fig 2. BMI for age z-scores (BAZ) among adolescent girls in Southern Ethiopia, 2019.**

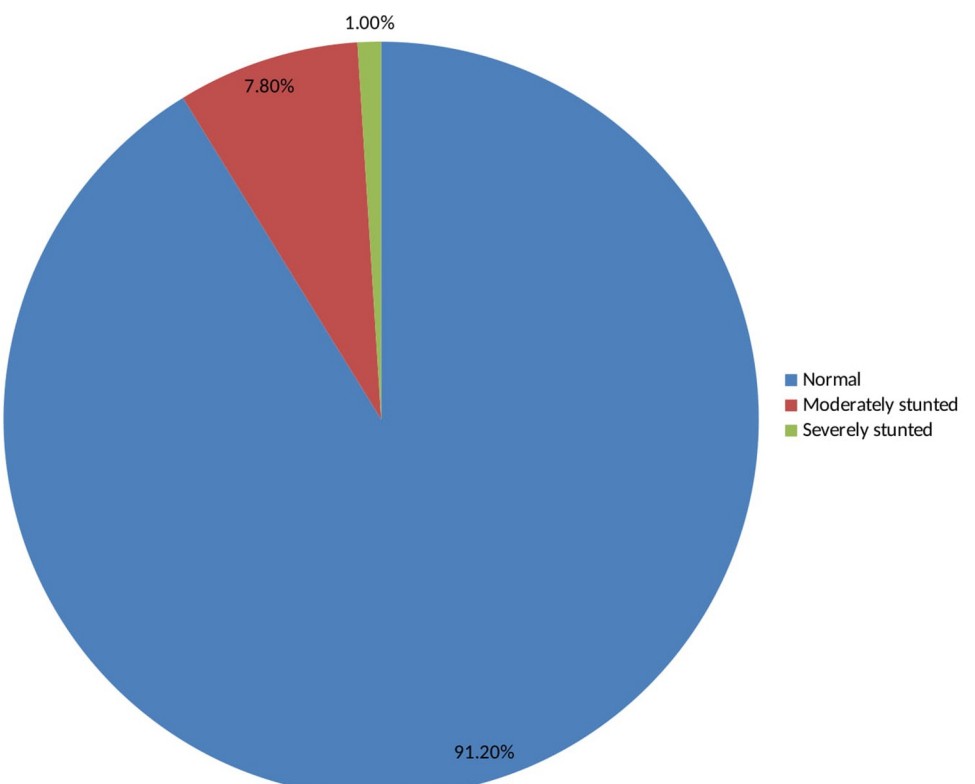

**Fig 3. Height for age z-scores (HAZ) among adolescent girls in Southern Ethiopia, 2019.**

## Discussion

### Health and nutritional status of adolescent girls in Southern Ethiopia

The BAZs revealed that 19.5% of the adolescent girls were moderately thin and 8% were severely thin. Similarly, 7.8% of adolescent girls are moderately stunted and 1% are severely stunted. The prevalence of thinness is higher in this study than in a study conducted in the Amhara Region [36]. Similarly, the prevalence of stunting in the current study was lower than in the study conducted at the Amhara Region [17] and in Adwa, northern Ethiopia [20]. The prevalence in this study were also lower than those reported from a study conducted in Bangladesh [19].

The reasons for the observed undernutrition among the current study participants might be due to their low monthly household income, large family size, low educational status of fathers' of the adolescent girls and poor hand-washing practice with soap before eating food. Our finding is supported by a study conducted in the Somali Region of Ethiopia, which indicated that hand washing with soap after using the toilet and before eating affects the nutritional status of adolescent girls [18]. This might be due to the association between diarrheal disease with not washing hands, which can also affect the nutritional status of adolescent girls [37]. Moreover, 40.4% of the study participants purchased their food from the market, whose amounts and quality can depend on their income, distance to markets, and price fluctuations [38].

The decision-making power of the family might also affect the nutritional status of adolescent girls. Decision-making for receiving nutrition services was found to be under the control of fathers' in 66.1% of the study participants. As indicated by a study conducted in

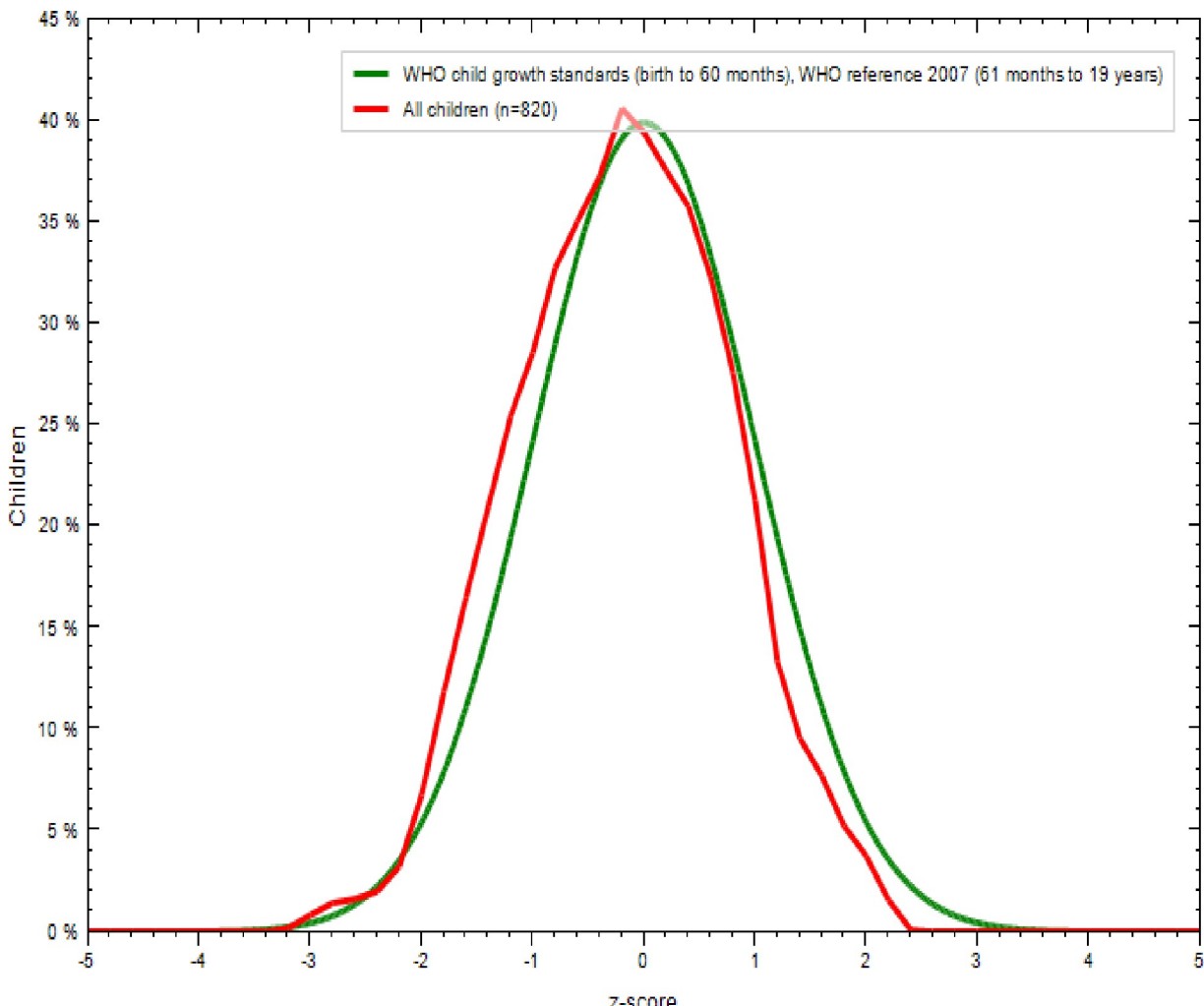

**Fig 4. Comparison of BMI-for-age z-scores (BAZ) of the study population (N = 820) with the 2007 WHO growth reference populations.**

Jimma Zone, Southwest of Ethiopia, women autonomy in decision-making is important to improve nutritional status women and their children [39, 40]. Similarly, 45.1% of the study participants did not receive a deworming tablet. As indicated by a study conducted in Angolela, Ethiopia, stunting of study participants was associated with intestinal parasite [41]. Therefore, this might further aggravate the low nutritional status of the study subjects [42].

### Factors associated with the nutritional status (BAZ) and HAZ of adolescent girls in Southern Ethiopia

Adolescent girls between the ages of 10–14 years were 2.9 times more likely to be thin than adolescent girls ≥ 15 years old. This finding is in line with the study conducted in the Amhara Region [17]. This might be due to the rapid growth and reproductive maturation during adolescence(10–14 years age), which increases energy and nutrient requirements and hence the need for quality diets [43, 44].

Adolescent girls with a family size > 5 were 1.6 times more likely to be thin than those from a family with ≤ 5 people. This finding is supported by studies conducted in the city of Arar

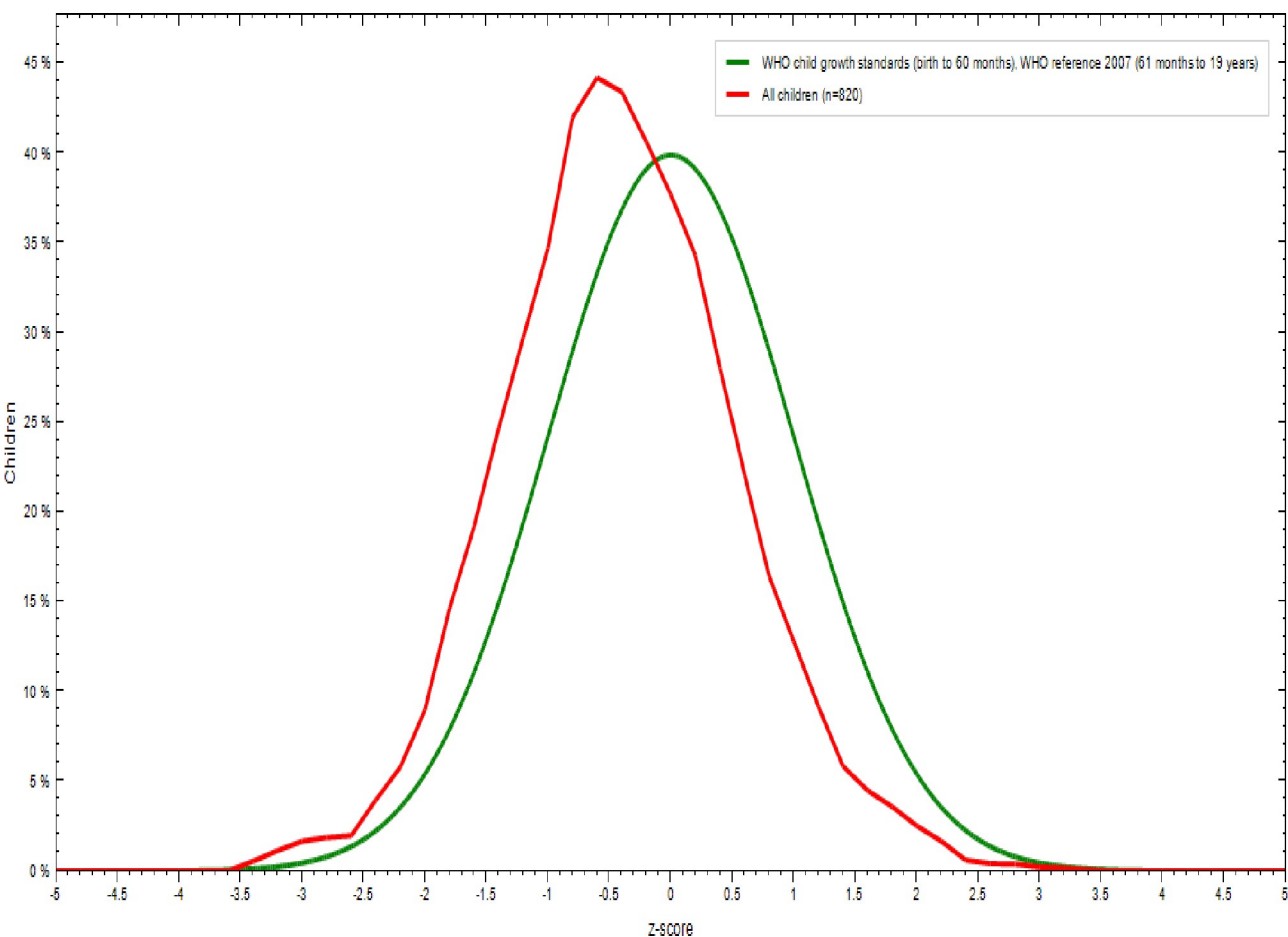

**Fig 5. Comparison of height-for-age z-scores (HAZ) of the study population (N = 820) with the 2007 WHO growth reference populations.**

[45], Nigeria [46], and the Amhara Region [17]. Large families may share food among family members [20], constraining the availability of adequate amount of quality food [47].

Adolescent girls from families whose monthly income was < 1000 ETB (31.25 USD) were 2.5 times more likely to be thinner than those from families who have monthly incomes > 2000ETB (62.5 USD). This finding is in line with studies conducted in Bangladesh [48, 49] and Nigeria [22]. This might be because the household's low monthly income, which affects the purchasing power leading to the consumption of suboptimal quantity and quality food. Consequently, adolescent girls from low monthly income were more likely to be thinner [50].

Study participants who did not take deworming tablets every six months were 1.56 times more likely to be thinner than those who took a deworming tablet every six months. According to the WHO preventive deworming recommendations, a biannual single-dose of albendazole (400 mg) is recommended as a public health intervention for all non-pregnant adolescent girls and women of reproductive age in order to reduce the burden of soil-transmitted helminthes which can affect nutritional status of adolescent girls [51]. Indeed, a systematic review and meta-analysis indicated that taking deworming tablets improves the nutritional status of adolescent girls [52], by causing diarrhea and reducing absorption of nutrients. The odds of stunting were also lower among adolescents that washed their hands prior to eating meals (P< 0.05).

**Table 6. Factors associated with nutritional status (BAZ) of adolescent girls in Southern Ethiopia, 2019.**

| Variables | | BAZ (≤ -2) N(%) | BAZ (> -2) N(%) | Crude OR (CI) | Adjusted OR (CI) |
|---|---|---|---|---|---|
| Age (years) | 10–14 | 144(17.6) | 249(30.3) | 2.397 (1.75–3.28)*** | 2.91 (2.03–4.17)*** |
| | > 15 | 83(10.1) | 344(42.0) | 1 | 1 |
| Family size | ≤ 5 | 57(6.9) | 195(23.8) | 1 | 1 |
| | > 5 | 170(20.7) | 398(48.6) | 1.46 (1.034–2.064)* | 1.63 (1.105–2.39)* |
| Monthly income ETB (USD) | < 31.25 | 126(15.4) | 148(18.1) | 3.37 (2.28–4.98)*** | 2.54 (1.66–3.87)*** |
| | 1000(31.25) -2000(62.5) | 49(6.0) | 249(30.3) | 0.779 (0.504–1.205) | 0.74 (0.475–1.158) |
| | > 2000(62.5) | 52(6.3) | 196(23.9) | 1 | 1 |
| | | Nutritional status | | | |
| Variables | | BAZ(≤ -2)N(%) | BAZ(> -2)N(%) | Crude OR (CI) | Adjusted OR (CI) |
| Receiving deworming tablets | Yes | 101(12.3) | 349(42.6) | 1 | 1 |
| | No | 126(15.4) | 244(29.7) | 1.8 (1.3–2.4)*** | 1.56 (1.1–21)* |
| Father's educational status | No formal education | 30(3.6) | 51(6.3) | 1.94 (1.1–3.4)* | 2.3 (1.1–4.8)* |
| | 1–8 grade | 73(8.8) | 190(23.2) | 1.28 (0.82–1.96) | 1.7 (0.96–2.87) |
| | 9–12 grade | 81(9.9) | 210 (25.6) | 1.27 (0.83–1.95) | 1.78 (0.86–3.01) |
| | College and University | 43(5.3) | 142(17.3) | 1 | 1 |
| Decision-maker for nutrition service | Father | 168(20.5) | 374(45.6) | 2.05 (1.37–3.07)** | 1.89 (1.22–2.94)** |
| | Mother | 23(2.8) | 55(6.7) | 1.905 (1.37–3.07)* | 2.022 (1.016–4.024)* |
| | Jointly | 36(4.4) | 164(20) | 1 | 1 |
| | | Nutritional status, N(%) | | | |
| Variables | | BAZ (≤ -2) | BAZ (> -2) | Crude OR (CI) | Adjusted OR (CI) |
| Visited by health extension worker regularly | Yes | 83(10.1) | 303(37.0) | 1 | 1 |
| | No | 144(17.5) | 290(35.4) | 1.81 (1.32–2.48)*** | 1.72 (1.7–2.4)** |
| Source of family food | Produce own | 109(13.3) | 306(37.3) | 3.74 (1.57–8.89)** | 3.288 (1.3–8.1)* |
| | Market purchase | 112(13.6) | 224(27.3) | 5.25 (2.21–12.5)*** | 5.14 (2.1–12.8)*** |
| | Produce own and market purchase | 6(0.73) | 63(7.7) | 1 | 1 |

*p-value < 0.05

**p-value < 0.01

***p-value<0.001

BAZ, BMI- for- age z-score

**Table 7. Factors associated with nutritional status (HAZ) of adolescent girls in Southern Ethiopia, 2019.**

| | | Nutritional status | | | |
|---|---|---|---|---|---|
| **Variables** | | **HAZ (≤ -2)** | **HAZ (> -2)** | **Crude OR (CI)** | **Adjusted OR (CI)** |
| Decision-maker for nutrition service | Father | 54(6.6) | 488(59.5) | 2.65 (1.241–5.68)** | 2.53 (1.106–6.087)* |
| | Mother | 10(1.2) | 68(8.3) | 3.529 (1.4–9.310)** | 2.58 (0.89–7.45) |
| | Jointly | 8(1.0) | 192(23.4) | 1 | 1 |
| Hand washing before eating and after toilet | Yes | 61(7.4) | 703(85.7) | 1 | 1 |
| | No | 11(1.4) | 45(5.5) | 2.82 (1.39–5.73)** | 2.25 (1.079–4.675)* |
| Visited by a health extension worker | Yes | 15(1.8) | 237(28.9) | 1 | 1 |
| | No | 57(7.0) | 511(62.3) | 2.13 (1.14–3.93)* | 2.036 (1.059–3.914)* |

*p-value < 0.05

**p-value < 0.01

***p-value<0.001

HAZ, height-for-age z-score

Study participants whose fathers had no formal education were 2.3 times more likely to be thinner than those whose fathers completed college and university level education. This finding is in line with a study conducted in the cities of Tehran [53] and in Adama in Central Ethiopia [54]. This might be due to educated families having better access to information, nutrition education, and quality diets [55].

Decision-making power for nutrition services was statistically associated with the nutritional status of the study participants. Adolescent girls that had both parents jointly making decisions on access to nutrition services were significantly less likely to be thin and stunted than adolescents whose decision to access to nutrition services were solely made by the father or the mother. Other studies have shown that women's participation in decision making is important for improving the nutritional status of women and children [56].

Adolescent girls acquiring their food from their household production or from purchases form the market were 3.28 and 5.14 times, respectively, more likely to be thinner than those who were getting their food from both home production and market purchases.

Visits by health extension workers were statistically associated with the nutritional status of adolescent girls. Adolescent girls who were not visited by health extension workers in their homes were 1.72 times more likely to be thinner than those who were visited by health extension workers at their homes within the past three months. This might be due to nutritional counseling that can result in the improvement of nutritional knowledge and behavioral change for improved nutrition [57]. Similarly, adolescents visited by health extension workers in the last three months had lower odds of being stunted.

## Conclusions

Thinness and stunting were found to be high in the study area. Age, family size, monthly household income, fathers' educational status, visits by health extension workers, and nutrition services decision-making power are the main predictors of thinness. Hand washing practice, visits by health extension workers, and nutrition services decision-making power are the main predictors of stunting among adolescent girls in Southern Ethiopia.

## Recommendation

- Income-generating activities should be implemented to improve the income of the families as it affects the nutritional status of adolescent girls.

- Health extension workers should visit and give nutrition education regularly for adolescent girls at their homes and at community meetings.

- Hand washing practice before meals and after visiting toilets should be improved

- Joint decision-making on household resources by both parents should be promoted

- Health extension workers should give counseling that discourages adolescent girls from skipping regular meals.

### Strength of the study

This study tried to include large sample size and relatively wider geographic area of the region which can be an input for the design and implementation of adolescent nutrition interventions.

### Weaknesses of the study

Using cross-sectional study design might not allow causal inferences. So, this study cannot tell cause and effect relationship.

## Supporting information

**S1 Checklist. STROBE statement.**
(DOCX)

**S1 File.**
(DOCX)

**S1 Data.**
(XLS)

## Acknowledgments

We acknowledge the Wolaita and Hadiya zones health office leaders and experts for their valuable cooperation during data collection. We would like to extend our gratitude to all the data collectors and adolescents who participated in this study. We are also grateful to the Center for Food Science and Nutrition, Addis Ababa University and Tufts University for the facilitation and support for the study.

## Author Contributions

**Conceptualization:** Yoseph Halala Handiso, Tefera Belachew, Abdulhalik Workicho, Kaleab Baye.

**Data curation:** Yoseph Halala Handiso, Cherinet Abuye, Abdulhalik Workicho, Kaleab Baye.

**Formal analysis:** Yoseph Halala Handiso, Tefera Belachew, Cherinet Abuye, Abdulhalik Workicho, Kaleab Baye.

**Funding acquisition:** Yoseph Halala Handiso, Abdulhalik Workicho, Kaleab Baye.

**Investigation:** Yoseph Halala Handiso, Tefera Belachew, Cherinet Abuye, Abdulhalik Workicho, Kaleab Baye.

**Methodology:** Yoseph Halala Handiso, Tefera Belachew, Cherinet Abuye, Abdulhalik Workicho, Kaleab Baye.

**Project administration:** Yoseph Halala Handiso, Tefera Belachew, Cherinet Abuye, Abdulhalik Workicho, Kaleab Baye.

**Resources:** Yoseph Halala Handiso, Tefera Belachew, Abdulhalik Workicho, Kaleab Baye.

**Software:** Yoseph Halala Handiso, Tefera Belachew, Abdulhalik Workicho, Kaleab Baye.

**Supervision:** Yoseph Halala Handiso, Tefera Belachew, Cherinet Abuye, Abdulhalik Workicho, Kaleab Baye.

**Validation:** Tefera Belachew, Cherinet Abuye, Kaleab Baye.

**Visualization:** Yoseph Halala Handiso, Tefera Belachew, Kaleab Baye.

**Writing – original draft:** Yoseph Halala Handiso, Tefera Belachew, Abdulhalik Workicho, Kaleab Baye.

**Writing – review & editing:** Yoseph Halala Handiso.

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
