## [Decision Letter · Decision Letter 0]

5 Mar 2020

PONE-D-19-32334

Under nutrition and its determinants among adolescent girls in low land area of southern Ethiopia

PLOS ONE

Dear Mr Handiso,

Thank you for submitting your manuscript to PLOS ONE. After careful consideration, we feel that it has merit but does not fully meet PLOS ONE’s publication criteria as it currently stands. Therefore, we invite you to submit a revised version of the manuscript that addresses the points raised during the review process.

The manuscript has been evaluated by two reviewers, and their comments are available below.

The reviewers have raised a number of concerns that need attention. In particular, please pay particular attention to the several suggestions left by Reviewer #2 (in the attached file) to improve the clarity of the reporting. Moreover, please also consider the comments of the first reviewer on the importance of considering age in your analysis. 

Could you please revise the manuscript to carefully address the concerns raised?

We would appreciate receiving your revised manuscript by Apr 19 2020 11:59PM. To enhance the reproducibility of your results, we recommend that if applicable you deposit your laboratory protocols in protocols.io, where a protocol can be assigned its own identifier (DOI) such that it can be cited independently in the future. For instructions see: http://journals.plos.org/plosone/s/submission-guidelines#loc-laboratory-protocols

We look forward to receiving your revised manuscript.

Kind regards,

Carmen Melatti

Associate Editor

PLOS ONE

Journal Requirements:

4. Please ensure that you refer to Figure 1-5 in your text as, if accepted, production will need this reference to link the reader to the figure.

5. We note that Figure 1 in your submission contains map images which may be copyrighted. All PLOS content is published under the Creative Commons Attribution License (CC BY 4.0), which means that the manuscript, images, and Supporting Information files will be freely available online, and any third party is permitted to access, download, copy, distribute, and use these materials in any way, even commercially, with proper attribution. For these reasons, we cannot publish previously copyrighted maps or satellite images created using proprietary data, such as Google software (Google Maps, Street View, and Earth). For more information, see our copyright guidelines: http://journals.plos.org/plosone/s/licenses-and-copyright.

Reviewers' comments:

Reviewer's Responses to Questions

**Comments to the Author**

1. Is the manuscript technically sound, and do the data support the conclusions?

Reviewer #1: Partly

Reviewer #2: Yes

2. Has the statistical analysis been performed appropriately and rigorously? 

Reviewer #1: N/A

Reviewer #2: Yes

3. Have the authors made all data underlying the findings in their manuscript fully available?

Reviewer #1: No

Reviewer #2: Yes

4. Is the manuscript presented in an intelligible fashion and written in standard English?

Reviewer #1: Yes

Reviewer #2: Yes

5. Review Comments to the Author

Reviewer #1: The manuscript reports on the nutritional status of adolescent girls in an area of southern Ethiopia and the association of socio-economic, health and environmental variables with normal versus low BMI z-scores (<-2.0 or >=-2.0) and normal versus low height-for-age (<-2.0 or >=-2.0). The study presents a reasonably large sample size (n=843) and employs univariate and multivariable logistic regression analysis to examine associations.

There are some limitations of the manuscript, namely: i. there is no specific research hypotheses that are being examined or tested; ii. there are several other similar published papers from Ethiopia examining factors associated with undernutrition and stunting in adolescent girls, so it is not apparent that this study adds any new or significant insights (see references number 18, 19, 21, 22)-other than being in a different region of the country; iii. this study does not include any novel approaches or methods that bring original or new insights into tackling nutritional challenges in the population; iv. the study does not consider the changes in BMI and height during the adolescent growth spurt and the timing of this relative to the growth spurt of the reference population used to calculate z-scores. Age should be included as a continuous variable in analyses to try to lesson the effects of age differences in adolescent growth; v) a whole range of variables are analysed without a specific rationale of how these relate to the research hypotheses.

The paper could also be improved by having more clearly defined research hypotheses, developing a conceptual framework based on detailed review of the literature and existing models or frameworks and aligning this with current government policies. The statistical analyses could use using model building to test these hypotheses based on the conceptual framework developed in the introduction section.

A more critical review of the literature, and a critical approach to the data analysis and interpretation (e.g. how do the data shed light on the effectiveness of current policies?, how reliable are self-reported data on hand-washing etc; is undernutrition really a priority health challenge if only 7.8% of adolescents experience stunting?) would provide a more valuable research contribution.

Overall, the weaknesses of the paper make it unsuitable for publication in its present form.

Reviewer #2: The manuscript requires thorough editing. The text to the tables that described the variables should be presented sequentially. The use of Nutritional status being associated with some parameters without stating the nature of the association is not proper.

6. PLOS authors have the option to publish the peer review history of their article (what does this mean?). If published, this will include your full peer review and any attached files.

Reviewer #1: No

Reviewer #2: Yes: RUFINA N.B. AYOGU

---

## [Author Response · Author response to Decision Letter 0]

18 Mar 2020

First of all, I would like to thank academic editor and reviewer#1 for their scientific, reasonable and convincing comments and questions. Really, I have got interesting lesson from them. Thank you very much. 

 I tarried to respond to all comments and questions bellow: 

Response for comments and questions raised by academic editor

1. We ensure that our manuscript meets PLOS ONE's style requirements. Our manuscript was edited and formatted by ‘editage Cactus’-licensed English edition. 

2. We have developed questionnaire after reading different literature based on objective of our study. We attached the questionnaire as supporting documents during manuscript submission. 

3. Yes, I have ORCID iD: https://orcid.org/0000-0001-6191-9027

4. We have referred to figure 1-5 in our text

5. All figures were not copied from other persons’ work. There are no copyright holders for these figures. These all figures are my own work. I have developed/ drawn these figures by using WHO Anthro plus, SPSS and ArcGIS software. 

Response for reviewer#1 comments and questions 

Abstract

Line 7: multistage random sampling if the authors used random in all the stages. 

Response: Yes, random sampling procedure was used. Equal chance was given all participants. 

Line 11: Did you actually estimate or generated through the software? 

Response: we generated through software by using SPSS version 21.0. 

Line 12: Did you actually estimate or generated through the software?

Response: we generated through software by using SPSS version 21.0. 

Line 18: Let us know the nature of the association. This is not clear.

Response: Nature of association is positive. 

Large family size, low monthly income , not taking deworming tablets , low educational status the fathers’, the source of food for the family only from market, not visited by health extension workers and not washing hand before eating and after using the toilet were positively associated with undernutrition of adolescent girls. 

Introduction

Line 24: Defined by who?

Response: age range of adolescent was defined by world health organization (WHO). 

Materials and methods 

Line 61: Materials and methods please.

Response: we changed it to “Materials and methods”

Line 66: Use an active verb such as shows instead of colon.

Response: we changed it to Figure 1 shows 

Line 71: BMI-for-age

Response: we changed it to BMI-for-age

Line 72: Height for age is a compound word and should be written as such.

Response: we changed it to height-for-age 

Line 73: Add level

Response: we added “level” 

Line 78: You did not use both (thinness and stunting) for sample size determination.

Response: we used both thinness and stunting for sample size determination. But, we took stunting because it gave larger sample size. 

Line 81: non response.

Response: we changed it to response 

Line 88: simple random sampling method. Was it by balloting or what?

Response: we inserted “sampling”; we used SPSS version 21.0 software for random selection. When we ran, it automatically selected 843 adolescent girls from all listed adolescent girls. First census was developed by health extension girls from each selected kebele. 

Line 92: Which type of random was used?

Response: If there was more than one adolescent girl in a household, one adolescent girl was randomly selected by lottery method. 

Line 93: Are kebele households? If no, how were households selected? There was no mention of house selection. What happened when there are more than one household per house? 

Response: No, kebeles are not households. Kebeles are “the lowest administrative unit in Ethiopia.”

Kebele includes many households. In average, 500-800 households are in one kebele. Census of adolescent girls was conducted from all randomly selected kebeles. So, all households in the randomly kebele which were with eligible adolescent girls were included in census.

Line 97: This is data collection method.

Response: we corrected it. 

Line 100: How? Were the authors the ones that calibrated the scale? Is calibration the same as maintaining the scale at 0?

Response: -No, the authors are not the ones that calibrate the scale. Data collectors calibrate the scale at each time whether it works properly or not. 

- Calibration is not only maintaining the scale at 0, but also it includes checking the tool is working correctly. 

Line 101: Was this calibrated or did you attach a measuring tape to it? This is not clear.

Response: we used the height measuring board which was with measuring tape. The measuring tape was attached to the board and fixed already. 

Line 102: BMI-for-age

Response: we corrected it 

Line 102: Height-for-age

Response: we corrected it 

Line 102: It would be better if the method of calculating the Z scores is explicitly shown.

Response: we used WHO Anthro-plus software to calculate Z- score from weight, height and age in month of adolescent girls.

Line 104: Was anthropometry not part of your data collection method?

Response: anthropometry is parts of data collection method. We corrected it 

Line 106: B.Sc.

Response: we corrected it 

Line 107: experiences in collecting

Response: we corrected it 

Line 109: M.Sc.

Response: we corrected it 

Line 110: Data were

Response: we corrected it 

Line 122: rendered or translated

Response: Back translation to English was conducted 

Line 125: This similar area needs to be mentioned.

Response: In similar kebeles which were selected for the study. 

Line 128: How many principal investigators did the study have?

Response: one, we corrected it. 

Line 131: When did you standardize the measurements? In the field or during training? This is not clear.

Response: Standardization was took place during training

Line 134: add (TEM)

Response: we added (TEM) 

Line 135: Let us have ENA in full first with the abbreviation in brackets.

Response: Emergency Nutrition Assessment (ENA)

Line 136: Were asked to instead of could.

Response: it was corrected 

Line 142: girls aged 

Response: it was corrected 

RESULTS: Variables in the tables should be reported in the text as the appeared in the tables i.e. sequentially. Results must be reported in past tense.

Line 149: This figure was not shown in Table 1. 

Response: we inserted/corrected the missed figure 

Line 150: This information is lacking in Table 1.

Response: we inserted/corrected the lacking information 

Line 150: This is not true of your Table 1.

Response: It was corrected 

Line 151: See Table 1 for clarity.

Response: It was corrected

Line 151: Table 1 does not have 5-8 nor 1-4.

Response: It was corrected 

Line 153: This figure is not the same as the figure on Table 1.

Response: It was corrected 

Line 153: figures are not in the table.

Response: It was corrected 

Line 153: Do not use about when you are stating the exact figure.

Response: It was corrected 

Line 154: It is good to also include this value in USD.

Response: <31.25 USD

Line 155: Let us have the value (>2000) in USD.

Response: >62.5 USD

Line 156: Use colon and not period i.e. Table 1:

Table 1

Response: we used colon 

Expunge level. 

Response: Level was expunged 

Line 172: Source: Field

Response: It was corrected 

 Line 174: Expunge approximately because you are stating the exact figure.

Response: the word approximately was expunged 

Line 177: Who took 

Response: it was corrected 

Line 177: 2 tablets of how many mg each?

Response: At this research study area, albendazole (400 mg) was given every six months for adolescent girls. But, adolescent girls don’t know the dose and types of deworming tablets. During data collection period, data collectors asked whether adolescent girls took deworming tablets or not. If they took, how many (one or two) deworming tablet. In the study area some adolescent girls were no taking deworming tablets. Because, they don’t know the importance of it. 

Line 178: This is a bit confusing. Why would they take two tablets every 6 months?

Response: in Ethiopia, adolescents were universally supplemented deworming tablets every six month. That means, twice annually adolescent girls were taking deworming tablets 

Line 179: This is not clear. What do you actually mean by having supplement/receiving supplements? Is this the same as taking it? Be specific on the type of supplements. There are quite a number of them

Response: that means taking iron-folic acid tablet 

Line 181: were not satisfied was not stated in Table 2. Either you include it in the table or you expunge it.

Response: It was expunged 

Line 185: Rearrange the variables so that they can be sequential. E.g. decision maker for nutrition service should be moved to after friendly nutrition service received.

On Table 2:

Response: it was corrected 

Expunge level. It was corrected. It was removed 

Is nutrition education not part of friendly nutrition service? 

Response: friendly nutrition education is parts of nutrition education.

It is how they are giving nutrition education politely. Friendly nutrition education is the way of approach how health extension giving nutrition education for adolescent girls. 

When? i.e. received nutrition education i.e. how long ago? 

Response: with in the last three months 

Received deworming tablets: How long ago? Duration needs to be stated.

Response: twice annually deworming tablets were given for adolescent girls. 

Friendly nutrition service given: use received instead of given.

Response: It was corrected 

Presence of cough, include 2 weeks before data collection.

Response: All are corrected by track change 

Line 189: The table says of adolescent girls and not of the study participants.

Response: corrected as ‘’adolescent girls’’ 

Line 190: See Table 3 for the correct figure.

Response: It was corrected 

Line 190: girls lived in houses with mud floors

Response: it was corrected 

Line 191: study participants lived with domestic animals.

Response: it was corrected 

Line 194: Figure (90.1%) is not the same as what is in the table.

On Table 3: 

Response: it was corrected as ‘’93.2%’’

Expunge level. 

Response: level expunged 

All sub percentages must add up to 100.0%

Response: it was corrected 

Recast as living with animals in the same house.

Response: it was corrected 

Number of windows in the entire house or per room? Be specific.

Response: It is the number of windows in the entire house 

Questions are not used as variables. 

Response: we have changed questions to variables 

The total percent is not up to 100.0%. It is 98.4%

Response: It was corrected 

Line 201: as indicated in Table 4, about 39.5%........

Response: It was corrected 

Line 201: No, it does not show that 60.5% skipped meals because 4 and above is not the standard. We have 3 standard meals a day. It rather shows propensity to overweight due to overeating.

Response: we corrected it. The right comment was given for us. We give many thanks for the reviewer#1. 

Line 206: 50.0% is not the same as the figure on the table.

Response: It was corrected. It is 50.6 % on the table 

Line 208: Meal pattern

Response: It was corrected 

Line 208: Why do you have a different table pattern here?

On Table 4

Expunge level.

Response: level expunged 

Meal skipped: Percentage should be based on 496 as 100.0%.

Response: It was corrected i.e. removed from the table 

Line 212: I believe that you mean -1 and median. Normal includes this and +1. So normal BMI-for-age refers to Z scores between -2 and +2 (excluding -2 and +2) or you say it ranges from -1 to +1.

Response: We used WHO 2010 manual for comparison and We used WHO Anthro plus software for Z-score calculation. 

 For this study: Normal, -2 < BAZ< +2

 Moderately thin, -3 ≤ BAZ < -2

 Severely thin, BAZ < -3

 Overweight, BAZ >+2. \\anthro_pc_manual_v322.pdf

Line 213: Moderate thinness or moderate stunting is BMI-for-age or Height-for-age Z score that equals -2 or you say less than (<) -1. 

Response: Moderately thin, -3 ≤ BAZ < -2 or moderately stunted -3 ≤ HAZ < -2

Line 214: Severe thinness or severe stunting is BMI-for-age or height-for-age Z score <2 or exactly 

-3.

Response: Z-score is < -3. 

Line 217 - 220: The figures should come in after Table 5.

Response: Now, I put figures after Table 5. 

Line 225: Details of the associations should be stated in all cases e.g. were the younger adolescents affected more than the older ones? Were children from households with family size of >5 affected more than those with 5 and less? 

Response: the details of the associations were now stated well.

Line 228: The variables should be discussed sequentially as they appeared in all the tables.

Response: It was corrected 

Line 232: The use of frequencies did not equalize the figures for comparison. I suggest the authors translate the N values to percentages for easy comparison.

Response: we translated frequencies to percentages 

On Table 6: BAZ is ≤-2

Response: It was corrected as BAZ is ≤-2

Line 234: Is this right? Do you not mean <0.01 for 0.001 and <0.001 for 0.000?

Response: it was corrected. We mean <0.01 for 0.001 and <0.001 for 0.000. Thank you very much

Discussion

Line 246: Restatement of results are not allowed. You can only refer to them.

Response: we corrected it

Line 252: than the findings/results from a study......

Response: we corrected it

Line 253: observed undernutrition among the……..

Response: we corrected it

Line 254 to 259 are on results. Results are not restated in discussion.

Response: now we have written it well. 

Line 260: indicated that hand washing with soap after using the toilet and before eating affects the nutritional status of adolescent girls. How? This should be made clear.

Response: This might be due to adolescent girls who were not washing their hand before eating and after using toilet can be affected by diarrhea and other communicable diseases. If they affected by diarrhea and other communicable diseases, absorption as well as utilization of nutrient can be affected, and which can lead to stunting. 

Line 261-262: This might lead to the low nutritional status of adolescent girls. Explain the mechanism through which this occurs.

This might be due to adolescent girls who were not washing their hand before eating and after using toilet can be affected by diarrhea and other diseases and which can lead to stunting.

Line 262: the result of a study conducted………

Response: It was corrected 

Line 264: replace was with as. 

Response: It was corrected 

Line 266: How does it affect their nutritional status?

Response: Buying food from distant market takes time and which can affect accessibility of the household for the food consumption. Sometimes market price fluctuation can affect food consumption. So, this might affect nutritional status of the adolescent girls. 

Line 269-271: Please state the implications of the results stated. 

As indicated by the study conducted in Jimma Zone and World Health Organization report, women autonomy in decision-making is important to improve nutritional status themselves and their children. 

Line 275 and throughout the discussion session, expunge malnourished and use only the specific terms: thinness or stunting. 

Response: we expunged malnutrition and used specific term thinness or stunting

Line 275: This is the association I expected to see in the result section. Please move it there and explain all associations there.

Response: now we corrected it well. 

Line 290-292: According to WHO preventive deworming recommendations, a biannual single-dose of albendazole (400 mg) or mebendazole (500 mg) is recommended. If this is so, why did you have 2 tablets?

Response: At this research study area, albendazole (400 mg) was given every six months for adolescent girls. But, adolescent girls don’t know the dose and types of deworming tablets. During data collection period, data collectors asked whether adolescent girls took deworming tablets or not. If they took, how many (one or two) deworming tablet. In the study area some adolescent girls were no taking deworming tablets. Because, they don’t know the importance of it. 

Line 293-294: What is the mechanism of action?

 Response: Intestinal parasite may share the food that adolescent girls have eaten. Sometimes, intestinal parasite cause diarrhea disease and which can decrease absorption of important nutrients. This can lead to low nutritional status of adolescent girls.

Lines 302-309: What you have here is result and not discussion.

Response: It was corrected 

Line 309: expunge malnourished and use thin.

Response: it was corrected. Malnourished was changed to specific word “thin”

Lines 324 to 331 were on results. Tell us the implications of your findings.

Response: now the implications are written well 

Line 325: ‘whose decision-maker the father or mother’ what is missing here?

Response: here is ‘’either…or’’ missing. we have corrected it. 

Acknowledgement

The use on ‘I’ indicates single author.

Response: it was corrected as “we”

Responses to the comments raised by Reviewer #1

i. there is no specific research hypotheses that are being examined or tested;

Response: there are specific research hypothesis that are stated well at the end of introduction section. 

Research hypothese1. Prevalence of under nutrition among adolescent girls in the study area is high

Research hypothesis2. Socio-economic, nutrition service, health and hygiene related variable are associated factors for under nutrition among adolescent girls

ii. there are several other similar published papers from Ethiopia examining factors associated with undernutrition and stunting in adolescent girls, so it is not apparent that this study adds any new or significant insights (see references number 18, 19, 21, 22)-other than being in a different region of the country;

 Response: study is new for southern Ethiopia among adolescent girls. 

iii. this study does not include any novel approaches or methods that bring original or new insights into tackling nutritional challenges in the population; 

Response: our study design is community based cross sectional, and which give equal chance to all adolescent girls to be included in the study. Most of the research conducted on school adolescent girls and which cannot be generalized for adolescent girls who are not following the school. But our research based on community. 

iv. the study does not consider the changes in BMI and height during the adolescent growth spurt and the timing of this relative to the growth spurt of the reference population used to calculate z-scores. Age should be included as a continuous variable in analyses to try to lesson the effects of age differences in adolescent growth; 

 Response: Age of the adolescent girls grouped as 1) Early adolescent (10-13years), 2) middle adolescents (14-16 years) and 3) late adolescents (17-19 years). WHO as well as many literatures grouped adolescent girls’ age like this. Growth spurt of adolescent girls is different in different age groups. Growth spurt of adolescent girls is high during 10-13 years old than 17-19years. During 10-13 years age, they need high quality and quantity diet. Analyzing age as continuous variable by linear regression model and analyzing age as grouped variable by binary logistic regression, there are no the effects of age differences in adolescent growth. So, our analysis by grouping age is not wrong. 

v) a whole range of variables are analyzed without a specific rationale of how these relate to the research hypotheses.

 Response: variables were analyzed with the specific rationale in relation to research hypotheses.

Vi) Developing a conceptual framework based on detailed review

I developed conceptual framework from literature review. Most of the time used for analysis but not submitted with manuscript.

---

## [Decision Letter · Decision Letter 1]

9 Jul 2020

PONE-D-19-32334R1

Under nutrition and its determinants among adolescent girls in low land area of southern Ethiopia

PLOS ONE

Dear Dr. Yoseph Halala Handiso

Thank you for submitting your manuscript to PLOS ONE. After careful consideration, we feel that it has merit but does not fully meet PLOS ONE’s publication criteria as it currently stands. Therefore, we invite you to submit a revised version of the manuscript that addresses the points raised during the review process.

ACADEMIC EDITOR: 

The authors are advised to ensure consistency and coherency in the methods section.  At one breathe,  they indicated that they used a census, however, they had a sample size and had a sampling approach. This inconistency must be corrected. Also, the authors should ensure that all the reviewers comments are addressed.

We look forward to receiving your revised manuscript.

Kind regards,

Eugene Kofuor Maafo Darteh, Ph.D.

Academic Editor

PLOS ONE

Reviewers' comments:

Reviewer's Responses to Questions

**Comments to the Author**

1. If the authors have adequately addressed your comments raised in a previous round of review and you feel that this manuscript is now acceptable for publication, you may indicate that here to bypass the “Comments to the Author” section, enter your conflict of interest statement in the “Confidential to Editor” section, and submit your "Accept" recommendation.

Reviewer #2: All comments have been addressed

Reviewer #3: (No Response)

2. Is the manuscript technically sound, and do the data support the conclusions?

Reviewer #2: Yes

Reviewer #3: Partly

3. Has the statistical analysis been performed appropriately and rigorously? 

Reviewer #2: Yes

Reviewer #3: Yes

4. Have the authors made all data underlying the findings in their manuscript fully available?

Reviewer #2: Yes

Reviewer #3: Yes

5. Is the manuscript presented in an intelligible fashion and written in standard English?

Reviewer #2: Yes

Reviewer #3: No

6. Review Comments to the Author

Reviewer #2: The authors have addressed most of the issues raised. The followings if addressed will put the manuscript in a better shape.

Abstract

Line 10: Replace estimated with generated.

Line 17: Please add poor to nutritional status.

Introduction

Line 23: Please reference WHO properly with the appropriate number.

Materials and methods

Line 93: was selected by simple random sampling (lottery method).

Line 108: B.Sc. Please add period after c.

Line 130: investigator

Results

Line 156: Please do not expunge your currency. Use your currency but insert USD in parenthesis for easy estimation of value e.g. > 2000 ETBirr (62.5 USD)

Table 1: Variables not variable

Table 1: Please see earlier comment. Let the USD value be in brackets. An example has been given.

N.B. All results must be presented in past tense.

See the example below:

Line 216 to 217: Report all results in past tense

E.g.

1. 48.7% of the study participants brushed their teeth.....

2. 53.3% washed........

3. 41.7% usually washed their........

4. 3.4% did not wash....... etc

Please results should be reported in past tenses.

Lines 233 and 238: Do Table 5 and figures 4 and 5 have the same information? I think the information they contain should be described differently.

Line 250: Low BAZ was statistically….

Table 6

Total for age vs BAZ ≤ -2 =227; BAZ > -2 = 587 = 814. What happened to 6 respondents?

Please do not expunge your currency. Use your currency but insert USD in parenthesis for easy estimation of value e.g. > 2000 ETBirr (62.5 USD)

Line 258: height-for-age

Line 259: Add low to HAZ of the study participants were statistically….

Table 7

Expunge level

HAZ ≤ -2 and not <-2

HAZ ≤ -2 =72 HAZ > -2 = 748 but Visited by a health extension worker has HAZ ≤ -2 =74 HAZ > -2 = 746. Which one is correct?

Discussion

Line 277: which can lead to diarrhoea.

Lines 304-305: Please do not expunge your currency. Use your currency but insert USD in parenthesis for easy estimation of value e.g. > 2000 ETBirr (62.5 USD).

Line 308: Families with lower monthly incomes are more likely to eat

Line 309: expunge that

N.B. I am not at peace with the tenses used in the result and discussion sections. The tenses should be reviewed. This is very important.

Reviewer #3: Thank you for the opportunity to review a revised manuscript entitled “Undernutrition and Its Determinants among Adolescent Girls in low land area of Southern Ethiopia”. Although I find the study interesting, and an improvement based on the previous comments, there are still major issues that need to be addressed to help improve the manuscript. These issues are outlined below:

Abstract

1. Line 14, should read low educational status of father, not status the father’s

2. This is a structured abstract, therefore I entreat the authors to separate their conclusion from the results section in the abstract

3. If the authors wish to caption the background of the abstract as background, then the introduction as found in the main manuscript must be changed to background or vice versa for consistency sake

Background

4. Replace developing countries with low and middle-income countries, line 25 page 1

5. Line 34 there is no full stop

6. Line 46, page 2, how large is large. Specify the exact proportion or percentage

7. Line 55 page 2, the sentence…is missing a connection. It presupposes the authors have reviewed previous studies conducted in the study area or creating a gap. As this is a great step, the gap is not clear/well-articulated. Please the gap very clear and strong.

8. The literature review on the predictors of under nutrition is not adequate. This needs strengthening. In addition, provide specific references for particular predictors. E.g low household income [2], age [3], type of place of residence [4] etc…

9. In addition, the predictors the authors found in their study were not reviewed at the background to inform the discussion appropriately

Materials and methods

10. Between line 60 and 61, please provide a sub-section “study area/setting”

11. Line 79 check the statement. It doesn’t read well recast

Sampling procedure

12. Line 87, p3 what informed the selection of the 2 zones? Likewise the selection of the of the three kebeles why 3?

13. Line 92, the authors stated that they used a census, why then did they use a lottery method to select one adolescent in household with more than one adolescent?

14. Again, if the authors claimed they used a census, why did they have a sample size?

15. Apart, from the issues raised, with regards to the census, the sampling approach the authors used is not well-articulated. This should be explained in detail.

16. Line 67-68, the authors mentioned that both in and out of school adolescents constituted the study sample. What was then the exclusion criteria? See line 96-97. Please describe those who were not eligible.

Data collection

17. Line 107 kindly make reference to the literature you adapted instrument/questionnaire from

18. Also make reference that the instrument is attached as a supplementary file.

19. Describe how the independent variables were measured. Especially monthly income. Why didn’t the authors use the approach by the demographic and health survey which uses household assets to create the wealth variables by using the principal component analysis technique

Data quality assurance

20. Specify the area the pretesting was done

Results

21. Check Table 1, the column with the percentages are not well presented

Discussion

22. Line 268, where is the reference for the study you are comparing your results with?

23. Line 276-277, please recast

24. Line 291, the independent variables should come before the outcome variable.

25. The authors failed to acknowledge the strength and weaknesses in their study. This should be discussed extensively

26. At the acknowledgement section, it appears the study was conducted by one author. This should be relooked at.

27. The authors should get a native English speaker to proofread the manuscript to correct errors.

7. PLOS authors have the option to publish the peer review history of their article (what does this mean?). If published, this will include your full peer review and any attached files.

Reviewer #2: **Yes: **DR RUFINA N.B. AYOGU

Reviewer #3: **Yes: **Abdul-Aziz Seidu

---

## [Author Response · Author response to Decision Letter 1]

27 Jul 2020

We responded for all Reviewer comments and questions

---

## [Decision Letter · Decision Letter 2]

15 Sep 2020

PONE-D-19-32334R2

Under nutrition and its determinants among adolescent girls in low land area of southern Ethiopia

PLOS ONE

Dear Dr. Yoseph Halala Handiso,

Thank you for submitting your manuscript to PLOS ONE. After careful consideration, we feel that it has merit but does not fully meet PLOS ONE’s publication criteria as it currently stands. Therefore, we invite you to submit a revised version of the manuscript that addresses the points raised during the review process.

I think there has been a marked improvement in the maunuscript. However, at the discussion section, the authors should limit the repetition of results."For example, "In this study, the BAZs of the study participants was statistically associated with the age of the adolescent girls (p<0.001)". Repetition the p-values at the discussion in my view is not necessary again. They are already at the results section.

The authors should kindly fill the STROBE CHECKLIST and attach it as a supplementary file.

We look forward to receiving your revised manuscript.

Kind regards,

Eugene Kofuor Maafo Darteh, Ph.D.

Academic Editor

PLOS ONE

Reviewers' comments:

Reviewer's Responses to Questions

**Comments to the Author**

1. If the authors have adequately addressed your comments raised in a previous round of review and you feel that this manuscript is now acceptable for publication, you may indicate that here to bypass the “Comments to the Author” section, enter your conflict of interest statement in the “Confidential to Editor” section, and submit your "Accept" recommendation.

Reviewer #2: All comments have been addressed

Reviewer #3: All comments have been addressed

2. Is the manuscript technically sound, and do the data support the conclusions?

Reviewer #2: Yes

Reviewer #3: Yes

3. Has the statistical analysis been performed appropriately and rigorously? 

Reviewer #2: Yes

Reviewer #3: Yes

4. Have the authors made all data underlying the findings in their manuscript fully available?

Reviewer #2: Yes

Reviewer #3: No

5. Is the manuscript presented in an intelligible fashion and written in standard English?

Reviewer #2: Yes

Reviewer #3: Yes

6. Review Comments to the Author

Reviewer #2: Variable in Table 3 should read variables.

With the exception of the above, the authors have satisfactorily responded to earlier queries raised.

Reviewer #3: Thanks to the authors for addressing most of my comments.

I must say the manuscript has improved substantially. Kudos!

However, at the discussion section, the authors should limit the repetition of results.

"For example, "In this study, the BAZs of the study participants was statistically associated with the age of the adolescent girls (p<0.001)". Repetition the p-values at the discussion in my view is not necessary again. They are already at the results section.

The authors should kindly fill the STROBE CHECKLIST and attach it as a supplementary file.

7. PLOS authors have the option to publish the peer review history of their article (what does this mean?). If published, this will include your full peer review and any attached files.

Reviewer #2: **Yes: **Dr Rufina N.B. Ayogu

Reviewer #3: **Yes: **Abdul-Aziz Seidu

---

## [Author Response · Author response to Decision Letter 2]

22 Sep 2020

Responses were given to all reviewers.

---

## [Editor Report · Decision Letter 3]

1 Oct 2020

Undernutrition and Its Determinants among Adolescent  Girls in low land area of Southern Ethiopia

PONE-D-19-32334R3

Dear Dr. Yoseph Halala Handiso

We’re pleased to inform you that your manuscript has been judged scientifically suitable for publication and will be formally accepted for publication once it meets all outstanding technical requirements.

Kind regards,

Eugene Kofuor Maafo Darteh, Ph.D.

Academic Editor

PLOS ONE
---

## [Editor Report · Acceptance letter]

14 Dec 2020

PONE-D-19-32334R3 

Undernutrition and Its Determinants among Adolescent Girls in low land area of Southern Ethiopia   

Dear Dr. Handiso:

I'm pleased to inform you that your manuscript has been deemed suitable for publication in PLOS ONE. Congratulations! Your manuscript is now with our production department. 

Kind regards, 

on behalf of

Dr. Eugene Kofuor Maafo Darteh 

Academic Editor

PLOS ONE